

# Harnessing the potential of chloroplast-derived expression elements for enhanced production of cellulases in *Escherichia coli*

Ayesha Siddiqui[1], Muhammad Mudassar Iqbal[1], Asad Ali[1], Iqra Fatima[1,2], Hazrat Ali[1], Aamir Shehzad[1], Sameer H. Qari[3], Ghulam Raza[1], Muhammad Aamer Mehmood[4], Peter J. Nixon[5] and Niaz Ahmad[1]

[1] National Institute for Biotechnology and Genetic Engineering College, Pakistan Institute of Engineering and Applied Sciences (NIBGE-C, PIEAS), Faisalabad, Pakistan
[2] Department of Biochemistry, School of Dental Medicine, University of Pennsylvania, Philadelphia, Pennsylvania, United States
[3] Biology Department, Al-Jumum University College, Umm Al-Qura University, Makkah, Saudi Arabia
[4] Department of Bioinformatics and Biotechnology, Government College University Faisalabad, Faisalabad, Pakistan
[5] Department of Life Sciences, Imperial College London, London, United Kingdom

Corresponding authors
Peter J. Nixon, p.nixon@imperial.ac.uk
Niaz Ahmad, niazbloch@yahoo.com

## ABSTRACT

Thermophilic cellulases can play a crucial part in the efficient breakdown of cellulose—a major component of lignocellulosic plant biomass, however, their commercial production needs simple and robust biomanufacturing biosystems. In this study, two cellulases (β-glucosidase and endoglucanase) were heterologously expressed in *Escherichia coli* under a chloroplast-derived constitutive promoter and expression-enhancing terminator. The genes encoding the cellulases were sourced from a thermophilic bacterium *Thermotoga maritima* to exploit their industrially needed thermotolerance potential. The codon-optimized gene sequences were synthesized and placed under a tobacco chloroplast 16S rRNA promoter (Prrn), along with the 5′ UTR (untranslated region) from gene 10 of phage T7 (T7g10). A six-residue long histidine tag (His$_6$-tag) was attached to the N-terminus for protein detection. A high-level of expression of β-glucosidase and endoglucanase in *E. coli* was recorded from the chloroplast promoter and terminator. Furthermore, the activity assays confirmed that the recombinant enzymes maintained their activity at elevated temperatures. Thermostability analysis showed that recombinant enzymes retained their thermotolerance even after being expressed in a non-native host. Where, β-glucosidase and endoglucanase showed their optimum activities at 90 °C and 100 °C, respectively. Examination of the 3D structures of *T. maritima* cellulases revealed differential ionic interactions contributing to this high degree of thermotolerance. The study highlights the feasibility of producing thermostable versions of recombinant enzymes in *E. coli* at high levels. Our finding underscores the potential of this approach to meet industrial demands for efficient enzyme production employing *E. coli* as a robust biomanufacturing platform.

## INTRODUCTION

The global shift towards sustainable energy solutions highlights the growing importance of non-food substrates to produce valuable, multifunctional materials (*Sufficiency et al., 2022*). Among these, biomass and renewable sources, particularly lignocellulose, are key in this transition due to their abundant and cost-effective nature (*Qamar & Pacifico, 2023*). Lignocellulose, rich in fermentable sugars, serve as renewable resources for biorefineries, producing high-quality chemicals and energy-efficient materials while minimizing environmental impact (*Bilal et al., 2018*). Lignocellulosic biomass, primarily comprising cellulose (40–50%), hemicellulose (15–25%), and lignin (~30%), constitutes about 90% of the plant's total composition. Cellulose is a linear polysaccharide composed of interlinked glucose molecules owing to strong β-1,4 glycosidic bonds, forming cellobiose units. In lignocellulosic biomass, the hydrogen bonds within cellulose form strong microfibrils, while hemicellulose adds to the structural complexity by incorporating a variety of polysaccharides, including galactan, xylan, and mannan. This intricate composition makes lignocellulose highly resistant to degradation.

Lignocellulose, while holding great promise for bioenergy applications, presents significant challenges due to its structural complexity, making it difficult to efficiently break down. Additionally, conventional chemical pre-treatment methods often degrade the material quality and contribute to environmental pollution (*Qamar & Pacifico, 2023*). Consequently, developing more sustainable and efficient approaches for lignocellulose processing is critical for advancing bioenergy technologies (*Jahangeer et al., 2024*). The conversion of lignocellulosic biomass into valuable products like biofuels involves three key stages: biomass pretreatment, enzymatic saccharification, and fermentation. Among these, enzymatic saccharification, which breaks down cellulose into fermentable sugars, is a crucial step. Its success largely depends on the activity and stability of the cellulolytic enzymes employed in the process.

Various microorganisms, such as fungi and bacteria, have evolved specialized enzymes that efficiently degrade cell wall components to obtain nutrients (*Sheldon & Woodley, 2018*). Microbial enzymes offer significant advantages as microorganisms can be rapidly cultured in large quantities on inexpensive media (*Nigam, 2013*) and are easier to recover, isolate, purify, and modify to improve performance (*Dhevagi et al., 2021*). Many enzymes require specific pH levels and temperatures for optimal activity, which often differ from those typical in industrial environments. However, extremophiles can produce enzymes that remain active under extreme conditions, such as high temperatures (above 70 °C), acidic or alkaline pH levels (pH < 3 or > 9), high salinity, and elevated pressure—that would denature standard enzymes. These enzymes are particularly valuable in industrial processes like biofuel production, chemical manufacturing, and food processing (*Gomes & Steiner, 2004*; *Gullo et al., 2006*; *Sujatha, Raju & Ramana, 2005*).

Cost-effective production of thermotolerant cellulases is essential for industrial applications, particularly in biofuel production from lignocellulosic plant biomass. The challenges have been the high costs associated with the production of enzymes processes (*Pihlajaniemi et al., 2021*), the enzyme stability and efficiency under the harsh conditions often required in industrial processes, such as elevated temperatures and acidic environments used during lignocellulosic biofuel pretreatment (*Yeoman et al., 2010*). Thermostable enzymes can accelerate reaction rates, increase substrate solubility, extend enzyme half-life, and reduce contamination risk, all of which contribute to lowering enzyme dosage and overall costs (*Yadav et al., 2018*; *Karnaouri et al., 2019*). *Thermotoga maritima* cellulases, such as β-glucosidase and endoglucanase, are particularly well-suited for these applications due to their exceptional thermostability and ability to function efficiently in low-pH environments. Their inclusion significantly enhances catalytic performance and cost-effectiveness, making them ideal for sustainable biofuel production and other industrial uses.

The heterologous yet cost-effective expression of cellulases has emerged as a key strategy to harness the superior properties of thermostable enzymes. Several studies have explored expressing cellulase and xylanase in bacteria, yeasts, fungi, and plants to boost thermostable enzyme production (*Juturu & Wu, 2012*; *Juturu & Wu, 2014*). While fungi have traditionally been used for cellulase production due to efficient extracellular enzyme secretion, bacterial systems have gained preference in recent years. Bacteria offer advantages such as faster growth, robustness under harsh conditions, and the ability to form multi-enzyme complexes, enhancing cellulase functionality and synergy (*Golgeri et al., 2024*). Additionally, bacteria can achieve high cell densities and yield substantial quantities of protein in a relatively short timeframe.

Our long-term objective is to express the express cellulases in higher plant chloroplasts. Chloroplast transformation offers several advantages, including the ability to compartmentalize enzymes within the organelle, preserving their structural integrity and functionality while minimizing degradation and proteolytic activity, thereby enhancing enzyme stability and activity (*Khan et al., 2018*; *Yu et al., 2020*). It is well-documented that transforming the chloroplast genome can lead to extraordinarily high expression levels (>72%; *Ruhlman et al., 2010*), which is highly desirable for the large-scale production of valuable targets for commercial applications. However, generating stable homoplasmic plants is a lengthy and labour-intensive process.

To expedite the evaluation of expression efficiency, this study represents a preliminary investigation aimed at assessing the performance of a chloroplast-derived expression cassette in a bacterial system. Given that chloroplast promoters and terminators are recognized by *E. coli's* genetic machinery, we sought to determine whether these elements could drive high levels of transgene expression in a bacterial system. Specifically, we selected the 16S ribosomal RNA operon promoter, Prrn, the strongest known promoter (*Yu et al., 2020*). Our strategy harnessed the high-expression capability of the chloroplast 16S rRNA operon promoter to produce thermostable enzymes in *E. coli*. While chloroplast expression elements are known to work in bacteria, they are not routinely employed for heterologous expression of recombinant enzymes in bacterial expression systems. This

approach combines the advantages of higher plant chloroplast gene expression elements, known for driving robust protein production, with the fast-growing and easy-to-culture bacterial system, allowing for efficient, large-scale enzyme expression. Both cellulase genes β-glucosidase and endoglucanase were expressed at high levels in their active form, with no loss to thermotolerance, demonstrating the fact that their stability was not compromised in a non-native system. This work demonstrates the effectiveness of our approach, utilizing chloroplast expression elements in bacterial system to create a scalable platform for recombinant enzyme production, tailored to meet industrial demands.

## MATERIALS AND METHODS

### Bacterial strains used and provided cell culturing conditions

*Escherichia coli* strains namely Top10 and/or DH5α were chosen for the purpose of cloning our target gene and construction of the vector plasmid owing to its very high and well reported transformation efficiency. The strains were propagated in the widely used Luria Bertani (LB) broth/agar (1% (w/v) tryptone, 0.5% (w/v) yeast extract, 1% (w/v) NaCl with addition of 1.5% (w/v) agar) for solid media at 37 °C. The *E. coli* strain BL-21 (DE3) was used for expression studies.

### Vector construction, transformation and identification of transformants

The nucleotide sequences encoding *T. maritima* β-glucosidase (EC 3.2.1.21; GenBank accession number: X74163.1) and endoglucanase (EC 3.2.1.4; GenBank accession number: CP004077.1) were retrieved from the NCBI GenBank database. Since the target enzymes are intended for expression in higher plant chloroplasts, to enhance their expression, codon optimization was performed to match the codon usage preferences of the chloroplast genome. This optimization was carried out using the Kazusa Codon Usage Database (https://www.kazusa.or.jp/codon/), ensuring improved translational efficiency in the target expression system.

For the construction of expression cassettes, we used the Golden Gate cloning strategy, with pKP9 serving as the Level 1 vector (*Zhou et al., 2008*). β-glucosidase and endoglucanase genes were cloned into this vector to create the Level 2 constructs. To facilitate their use in the Golden Gate system, both the target genes and vector backbone were modified to incorporate PaqCI restriction sites using a set of primers (Table S1). This system utilizes Type IIS restriction enzymes, which generate non-palindromic overhangs, allowing for directional and order-specific assembly of DNA fragments. This approach significantly minimizes the risk of insert misorientation and enhances cloning accuracy (*Engler, Kandzia & Marillonnet, 2008*). The cloning vector, pKP9, was provided by Prof. Ralph Bock at the Max Planck Institute of Molecular Plant Physiology, Potsdam, Germany (*Zhou et al., 2008*). Briefly, the coding regions were linked to a strong constitutive 16S ribosomal RNA promoter, Prrn. A six-histidine tag (His$_6$-tag) was added to N-terminus of expressed sequences for protein detection and quantification. A terminator of *rps16* (Trps16) was used to halt the ongoing transcription *via* polymerase and stabilize the encrypted strands (Fig. 1 for details).

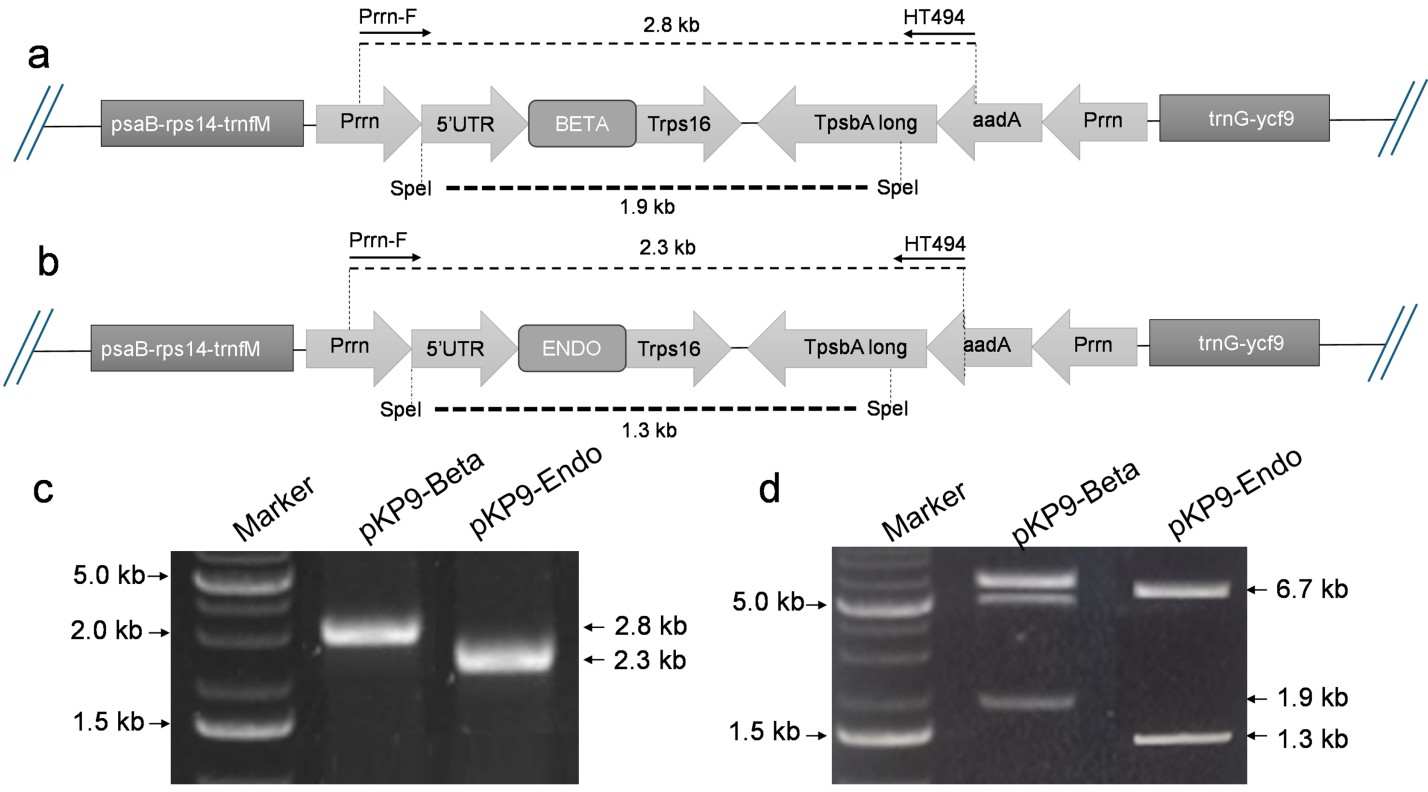

**Figure 1 Construction of expression vectors.** (A) and (B) show the physical map of the constructs, annealing sites of the primers and the restriction site of SpeI enzyme for digestion. The dotted lines indicate the expected fragment size to be amplified or released. (C) Shows the gel electrophoresis of fragment of interests amplified using PCR. (D) shows the gel electrophoresis of the fragments released after digestion with SpeI. The amplified or released fragments are displayed on basis of size (in kilo-bases) on the right-hand side of the gels, while the marker fragment sizes used for calibration are shown on the left. A 1% (w/v) gel was used and stained with ethidium bromide. The gel was calibrated using 1 Kb DNA Ladder (Thermo Fisher Scientific, Waltham, MA, USA).

The Golden Gate reaction was set up in a total volume of 10 μl, consisting of 3 nM of each insert, 1 nM of vector DNA, 1 μl of 10× ligase buffer, 0.5 μl of T4 DNA ligase, 0.5 μl of Type IIS restriction enzyme PaqCI, and 7.15 μl of dH$_2$O. The Golden Gate assembly protocol was as follows: an initial incubation at 37 °C for 15 min, followed by 30 cycles of 37 °C for 5 min and 16 °C for 5 min. This was then followed by a final incubation at 37 °C for 15 min and 65 °C for 20 min. The *E. coli* strain DH5α was transformed with the ligation mixture and plated on LB agar. Bacterial transformants were selected on 100 μg/ml ampicillin/carbenicillin and screened by colony polymerase chain reaction (PCR) using Prrn-F (5′-AGCTCGGTACCCCAAAGCTCCCCCG-3′) as forward primer and HT494 (5′-TACATTTGTACGGCTCCG-3′) as a reverse primer. The resulting expression vectors pKP9-BETA and pKP9-ENDO were confirmed through enzyme digestion, PCR, and sequencing. PCR amplification was conducted using specific primer sets to confirm each construct. For pKP9-BETA, confirmation was achieved using the primers BETA-For (5′-CAGGAAGGGTGAATCAGAAA-3′) and BETA-Rev (5′-TTGAGTATTCTGCGAACCAA-3′). Similarly, pKP9-ENDO was confirmed with the primers ENDO-For (5′-GACTGAGGCATCTATTGGAG-3′) and ENDO-Rev

(5′-CGAGCCATAATTCCCATGTA-3′). DNA integrity and successful modifications were verified on 1% (w/v) agarose gel electrophoresis, stained with ethidium bromide/ SYBR safe, and visualized under UV light.

## Protein extraction, SDS-PAGE and western blot analysis

Both the transformed and untransformed cells were cultured in 10 ml of LB broth and incubated at 37 °C for 16–18 h until the $OD_{600}$ reached 0.4–0.6. Subsequently, 3 ml of the bacterial culture was centrifuged at 14,000 rpm at 4 °C for 2 min. Following centrifugation, the separated supernatant was saved to analyse extracellular proteins, and the cell pellet was resuspended in 150 µl of 150 mM Tris-HCl buffer (pH 7.5), with the addition of 1.5 µl of 100 mM PMSF and 15 µl of 10 mg/mL lysozyme. The cell pellet was thoroughly mixed by vortexing. The microfuge tube containing the cellular lysate was then incubated on ice for 30 min, gently thawed and mixed by vortexing every 5 min. After 30 min, the lysate was centrifuged at 10,000 rpm, 4 °C for 10 min, resulting in the separation of two protein fractions. The supernatant, containing soluble proteins, was transferred to a new sterile microfuge tube and kept at 4 °C. The pellet, containing insoluble proteins, was resuspended in 1.5 ml of 150 mM Tris-HCl buffer (pH 7.5) by vortexing and stored at 4 °C. Protein quantification was carried out using Bradford assay (*Bradford, 1976*).

To visualise the expression of targeted enzymes in *E. coli* cells, extracted proteins were separated by SDS-PAGE using 12% (w/v) gels, and visualised by staining with Coomassie blue. After the proteins were separated on gel, they were transferred (or "blotted") onto a nitrocellulose membrane and incubated at 4 °C overnight before incubation with secondary antibody. The membrane was then probed with anti-His antibodies at 1:3,000 dilution to detect the target protein. The secondary antibody used was anti-mouse IgG alkaline phosphatase conjugate (Sigma, St. Louis, MO, USA) at a dilution of 1:10,000. Alkaline phosphatase substrate (Thermo Fisher Scientific, Waltham, MA, USA, 1 Step™ NBT/BCIP) was used for visualizing the bands.

## Enzyme activity assays

### β-glucosidase enzyme activity

Activity of expressed β-glucosidase was qualitatively analysed using cellobiose plate assay. 1% cellobiose-agar mixture (1 g in 100 ml $dH_2O$, 1.5 g agar to solidify) was transferred into Petri plates and allowed to set, creating a solid growth substrate and wells were created to inoculate protein isolated from *E. coli* cells. After the inoculation of enzyme samples (0.1 ml of extracts were used) plates were incubated at 40 °C for 24 h, following negative staining with Congo red conducted to the plates. The plates were then left to allow the Congo red to bind to the cellobiose substrate and washed with 1 M NaCl and left for 5 min to visualize zones due to hydrolysis caused by β-glucosidase activity.

### pNPG assay for β-glucosidase quantification

The pNPG (p-nitrophenyl β-D-glucopyranoside) method is a widely used initial reaction rate assay that allows for the measurement of enzyme activity (*Rajoka & Malik, 1997*). To

conduct the assay, 1.0 ml of pNPG solution (5 mM) and 1.8 ml of 0.1 M acetate buffer pH 4.8 was added to test tubes and equilibrated at different temperatures (50–110 °C). 0.2 ml of the enzyme solutions were added to tubes containing the substrate, followed by thorough mixing. Enzyme blanks and substrate blanks were also prepared by adding the appropriate solutions to separate tubes. All tubes are then incubated at different temperature for 30 min. The reaction was halted by adding 4 ml glycine buffer (0.4 M pH 10.8). The absorbance of the liberated p-nitrophenol product was determined at 410 nM, with readings compared to the substrate blank. The enzyme activity was measured by calculating the amount of p-nitrophenol released.

All enzyme assays were conducted in triplicate. Enzyme activity was quantified using a calibration curve for pNP. One unit (U) of enzyme activity is defined as the amount of enzyme that releases 1 μmol of pNP per minute at a temperature of 90 °C.

### Enzyme activity assay for endoglucanase activity

The activity of endoglucanase enzyme was analysed using a carboxymethyl cellulose (CMC) plate assay as reported by *Ghose (1987)*. For this assay, 1% (w/v) CMC agar plates were prepared. The prepared agar mixture was poured into petri dishes, and wells were created to inoculate protein from *E. coli* cells transformed with or without pKP9-ENDO. After the inoculation of enzyme samples (0.1 ml of extracts were used) plates were incubated at 40 °C for 24 h, followed by negative staining with Congo red was conducted to the surface of the plates. The plates were then incubated for 20 min to allow the Congo red to bind to the CMC substrate and washed with 1M NaCl for 5 min to visualize zones of hydrolysis produced due to endoglucanase activity.

### Enzyme assay for endoglucanase quantification

The DNS (3,5-dinitrosalicylic acid) method (*Wood et al., 2012*) is utilized to quantify the level of reducing sugars in a sample. For this assay, 0.1 mL enzyme sample was combined with a 1% CMC solution in 1 mL of 50 mM sodium acetate buffer pH 4.0 and incubated at varying temperatures (ranging from 50 °C to 110 °C) in a shaking water bath. The reaction was halted by boiling for 5 min followed by cooling for 20 min. The reaction mixture now called Quench Reaction Mixture (QRM) (200 μl) was used for estimation of reducing sugar produced by addition of 2 ml DNS reagent, and the reaction mixture after vortexing was boiled for 10 min. Enzyme activity was determined by measuring the absorbance change at 550 nM and relating this to the amount of glucose released using a glucose calibration curve.

All cellulase assays were carried out in triplicate. One unit (U) of enzyme activity is defined as the quantity of enzyme required to release 1 μmol of glucose per minute at a temperature of 90 °C.

## Thermotolerance assays

To assess the thermostability of β-glucosidase and endoglucanase enzymes, the isolated enzyme samples were divided into microcentrifuge tubes and subjected to their respective determined optimum temperatures using a thermal cycler or incubator. Incubation was carried out every 5 min interval for up to 60 min to monitor the

enzyme activity. Enzyme activity assays were conducted as described in previous section (See pNPG assay for β-glucosidase quantification and Enzyme assay for endoglucanase quantification). The enzyme activity was determined using a spectrophotometer, and the data was analysed to generate a First Order Plot by plotting enzyme activity against time for both enzymes.

## Kinetics of substrate hydrolysis

The Michaelis-Menten kinetic constants ($V_{max}$ and $K_m$) for β-glucosidase and endoglucanase were determined by assaying the enzymes on varied concentrations of substrates (pNPG & CMC) as described earlier. The kinetic constants for substrate hydrolysis were determined by direct fit method (non-linear regression). Briefly, kinetic constants of β-glucosidase for pNPG hydrolysis at 90 °C, pH 4.8 were determined by incubating the fixed amount of β-glucosidase extract (27.85 μg) with varied concentrations of pNPG ranging from 0.02 to 1.0 mM as described (*Javed et al., 2009*). Stock solution of pNPG (5.0 mM) was prepared in 50 mM sodium acetate buffer, pH 4.8 and required concentrations of pNPG were prepared by diluting the stock with the buffer.

The endoglucanase was assayed on varied CMC concentrations, ranging from 0.05–3.0% (w/v), which were prepared by using CMC stock solution (5% w/v). The endoglucanase activity was performed at 90 °C, pH 4.0 and fixed amount of endoglucanase extract (22.53 μg) was used per assay. The required concentration was achieved by using the following equation:

$$M_1V_1 = M_2V_2$$

where, $M_1$ = stock concentration, $M_2$ = required concentration, $V_1$ = volume of stock solution, $V_2$ = volume of solution to be made.

## Statistical analyses

The data obtained were subjected to various statistical analyses utilizing R Environment 4.1.3 for Windows. The data for enzyme activity were tested for significance using one–way analysis of variance (ANOVA) keeping $\alpha = 0.05$ followed by Tukey's Honest Significant Difference (HSD) *post-hoc* test for pairwise comparison of means between groups.

## Structural analysis of enzymes for thermostability

Three–dimensional (3D) structures of endoglucanase (PDB codes: 3AMH and 3AMM, *Cheng et al., 2011*) and β-glucosidase (PDB codes: 1UZ1, *Vincent et al., 2004*; 2CBU, *Gloster, Madsen & Davies, 2006*; 2WBG, *Aguilar-Moncayo et al., 2009*) from *T. maritima* were accessed from the Protein Data Bank (PDB) (*Berman et al., 2000*). Structural features relevant to thermostability of the afore-mentioned enzymes were compared with that found in homologous structures from thermophilic and mesophilic organisms. PyMOL (The PyMOL Molecular Graphics System, Version 1.3, Schrödinger, LLC) was used for visualizing 3D structures.

## RESULTS

### Confirmation of expression vectors based on chloroplast-derived expression elements

To express β-glucosidase and endoglucanase in *E. coli* from the chloroplast gene expression elements (promoter, untranslated regions (UTRs) and terminator), the tobacco chloroplast expression vector pKP9 was used. This vector contains the strongest known promoter in the chloroplast genome, 16S rRNA Promoter-Prrn and a Trps16 terminator. The coding sequences of β-glucosidase and endoglucanase were cloned in pKP9 as PaqCI fragments. In addition, a hexa-histidine-tag (His$_6$-tag) was N-terminally fused for identification through immunoblotting (Fig. 1). The constructed plasmids pKP9-ENDO and pKP9-BETA were restricted using SpeI enzyme, resulting in the release of expected fragment sizes of 1.9 and 1.3 kbp, respectively (Fig. 1). In addition to restriction digestion, the constructs were also validated through PCR using gene specific primers. The amplification of desired length of fragments confirmed the correct integration of the transgenes at the desired position (Fig. 2).

### Enhanced protein expression in *E. coli*

The recombinant vectors after being constructed and confirmed to contain the desired sequences were then transformed in commercially available *E. coli* strain BL21(DE3). Upon reaching an OD$_{600}$ between 0.4 and 0.6, the soluble and insoluble proteins were isolated both from BL21(DE3)-BETA (BL21(DE3) cells expressing β-glucosidase), BL21 (DE3)-ENDO (BL21(DE3) cells expressing endoglucanase) and untransformed cells and separated through SDS–PAGE. Upon staining with Coomassie Brilliant Blue, an additional fragment equivalent to the theoretical size of β-glucosidase (~52 kDa) and endoglucanase (~30 kDa) were observed in the soluble fraction of transformed cells while no additional band could be detected in any fraction of the untransformed cells (Fig. 3A). To confirm the identity of the additional fragments observed, the resolved proteins on the gel were then carefully transferred to a nylon-based membrane sheet and probed with anti-His antibodies. This immunoblotting confirmed the accumulating proteins were β-glucosidase and endoglucanase. The detection of enzyme in the insoluble fraction as a weaker band suggests that the proteins were predominately accumulating in the soluble fractions (Fig. 3B).

### Physiological activity of β-glucosidase

After confirmation of accumulation of recombinant enzymes in bacteria, we then determined whether the expressed proteins are physiologically active as well. The activity of β-glucosidase was measured due to its sensitivity and specificity. The pNPG assay is a widely accepted method for quantifying β-glucosidase activity because it provides a clear and quantifiable colorimetric readout that directly correlates with enzyme activity. Upon hydrolysis of pNPG by β-glucosidase, p-nitrophenol is released, producing a yellow colour measurable at 410 nM. This method allows for precise kinetic analyses and is highly effective in assessing the enzyme's activity in various conditions (*Rajoka & Malik, 1997*).

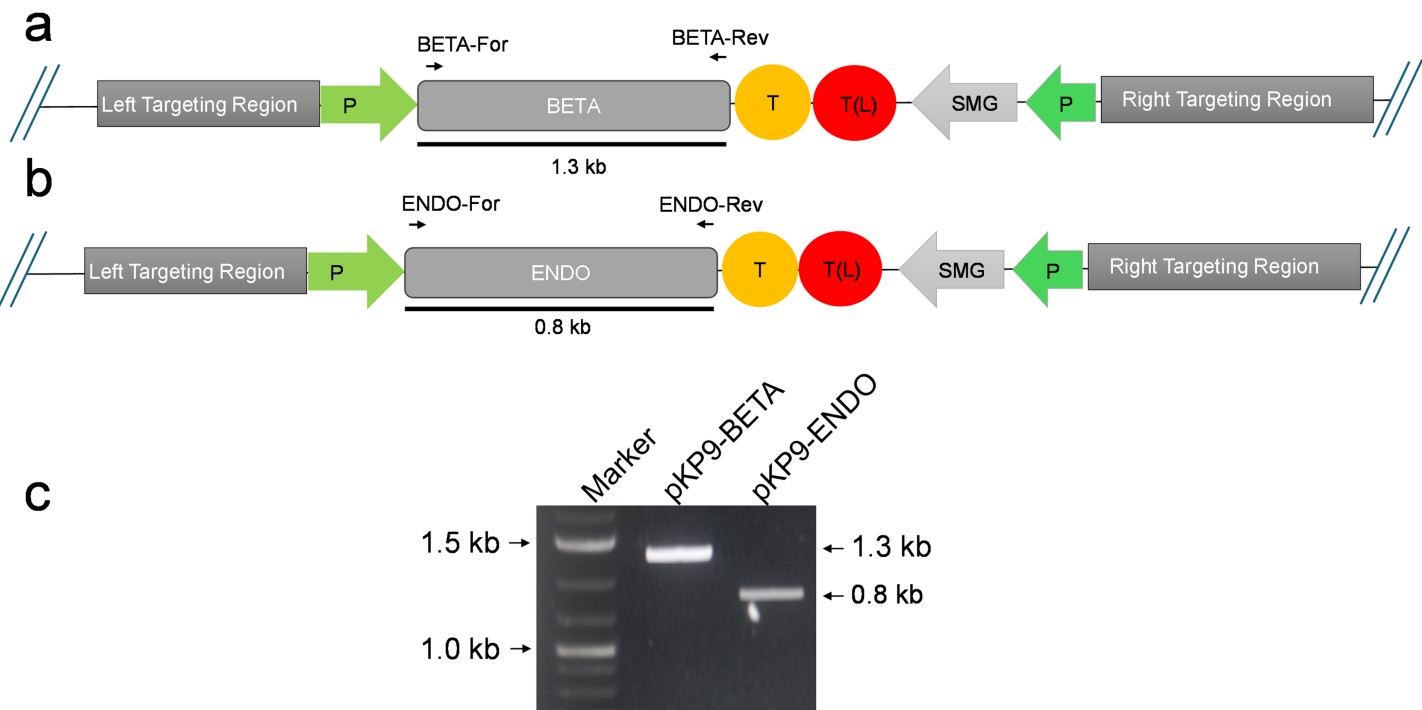

**Figure 2 Confirmation of *E. coli* transformants through PCR.** (A) and (B) show the physical map of the construct carrying the gene of interest (GOI) for expression in *E. coli.* The position of the primers is shown by small arrows while thick line indicates the fragment to be amplified. (C) shows gel electrophoresis of amplicons from *E. coli* transformants. The amplicons were analysed on a 1% (w/v) agarose-based gel electrophoresis. The fragments were standardized using 1 Kb DNA marker or ladder (Thermo Fisher Scientific, Waltham, MA, USA). The expected size of fragments (in kilo-bases) are indicated on the right-hand side, while the marker fragment sizes are shown on the left of gel image. P, promoter; T, terminator; SMG, selectable marker gene, T(L), terminator long.

The pNPG assay revealed that the enzyme was active in both the soluble and insoluble fractions of intracellular extracts, with no enzymatic activity detected in control cells. This enzymatic activity was evidenced by a colorimetric change, indicative of the enzyme's capacity to hydrolyze p-nitrophenyl-β-D-glucopyranoside (Fig. S1). Quantitative assay results manifested a significant increase in absorbance at 410 nM, consistent with the enzymatic release of p-nitrophenol. These findings demonstrate the enzyme's functionality across various intracellular compartments (*Strahsburger et al., 2017*).

## Absence of endoglucanase secretion into the extracellular medium

For endoglucanase, we utilized the carboxymethyl cellulose (CMC) plate assay, which is particularly suitable for detecting and visualizing endoglucanase activity on a solid substrate. The CMC plate assay involves the application of the enzyme onto a plate containing CMC embedded in an agarose gel. Upon incubation, areas where endoglucanase has hydrolyzed the CMC become clear against a dark background when stained with Congo red. This assay is chosen for its ability to provide a spatial visualization of enzyme activity, which is crucial for comparative analyses of enzyme action across different samples. Additionally, it is a straightforward and cost-effective method to assess

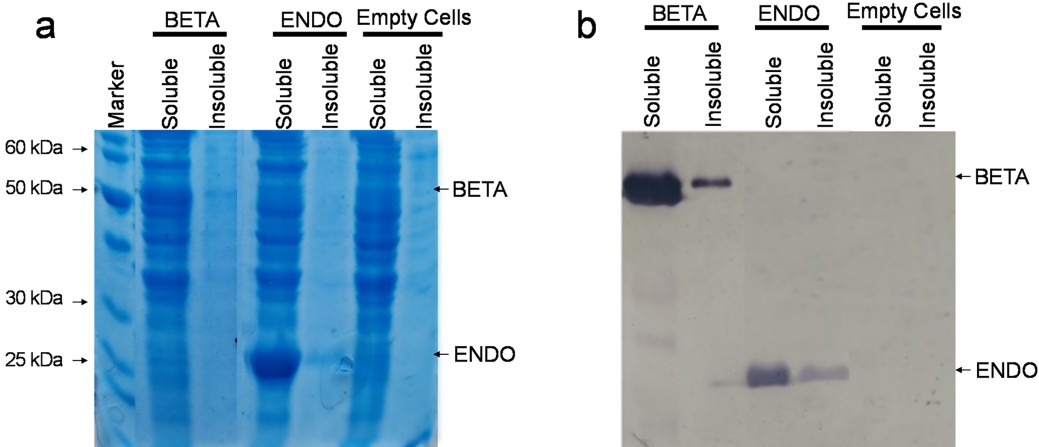

**Figure 3  Detection of recombinant proteins by SDS–PAGE and immunoblotting.** (A) SDS–PAGE of proteins isolated from *E. coli*. The total proteinaceous fraction was extracted from these *E. coli* cells transformed containing pKP9-BETA and pKP9-ENDO vector constructs as well as empty (vector-free) cells. Soluble and insoluble fractions were separated on a 12.5% (w/v) denaturing polyacrylamide gel. After quantification, 0.13 μg proteins were loaded in each well. The size of proteins was calibrated using a 200 kDa pre-stained Protein Ladder (Thermo Fisher Scientific, Waltham, MA, USA). The size of the fragments of the protein marker is shown on the left handside of the gel. (B) Immunoblotting of the total protein fraction from *E. coli*. Soluble and insoluble protein fractions were extracted for analysis from *E. coli* cells expressing betaglucosidase and endoglucanase, it was then transferred to a nitrocellulose membrane and probed with monoclonal anti-polyHistidine antibodies.

the activity of endoglucanases, particularly suitable for preliminary screenings in our experimental design.

The clear zones formed due to hydrolysis around the wells on the agar plates showed that expressed endoglucanase is in an active conformation (Fig. 4). The absence of halozones in extracellular extract suggests that the enzyme was not secreted in the medium. Both the soluble and insoluble fractions of intracellular extract showed visible activity by forming halozones. The presence of these clear zones indicates that our enzyme produced is active and can actively degrade CMC substrate, leading to the formation of soluble sugars and a visible hydrolysis pattern.

## Optima temperature requirement of recombinant enzymes

Optimum temperatures of recombinant β-glucosidase were determined by testing enzymatic activity from 50 °C to 110 °C using the pNPG assay. The enzyme showed optimal activity at 90 °C (Figs. 5A, 5B). Earlier studies on the heterologous expression of *T. maritima* β-glucosidase in *E. coli* reported an optimal activity at pH of 5.0–7.0 and at temperatures from 80–100 °C (*Mehmood et al., 2014*).

Likewise, the degree of thermotolerance for endoglucanase was determined using a DNS assay. The enzyme showed optimal activity at 100 °C whereas further increase in temperature showed slight change in activity (Figs. 5C, 5D). The optimal temperature requirement for recombinant endoglucanase was in line with earlier results (*Bok, Yernool & Eveleigh, 1998*).

## Thermostability of recombinant enzymes

To examine thermal stability, the enzymes were preheated at their respective optimum temperature for 1 h at an interval of 5 min. Both the enzymes were found to be stable after heating and showed significant activity even after 60 min incubation (Fig. 6). In the analysis of the enzyme-catalysed hydrolysis of pNPG/CMC using the pseudo-first-order kinetics model, a plot of the natural logarithm of the enzyme activity against time (t) was generated. This approach allows for the quantification of enzymatic reaction rates and determining rate constant (k) governing the reaction kinetics.

Both enzymes showed excellent thermal stability after treatment at high temperature above 80 °C. The graph showed that the enzymes β-glucosidase and endoglucanase retained 85% ($P = 2.2 \times 10^{-4}$) and 77% ($P = 6.4 \times 10^{-6}$) of activity, exhibiting a half-life of 123 and 167 min, respectively.

## Enzyme kinetics of recombinant β-glucosidase and endoglucanase

The kinetics of pNPG hydrolysis by β-glucosidase at 90 °C, pH 4.8 were determined by applying the direct fit model (Fig. 7A). The specific activity of intra cellular β-glucosidase for pNPG hydrolysis by *T. maritima* gene expressed in *E. coli* was 283.2 U mg$^{-1}$(Table 1). The Michaelis Menten constant ($K_m$) for pNPG was 0.136 mM. The expression protocol used in our study was very efficient as compared to the previous report, which deals with the heterologous expression of β-glucosidase gene (*bgl*A) from *T. maritima* in *E. coli* M15 (*Mehmood et al., 2014*). However, in their report the kinetic constants for pNPG hydrolysis were determined at 80 °C. We found that the affinity of pNPG to the active site ($K_m$) was about 4-fold higher, which confirmed that the pNPG was highly specific to our β-glucosidase as compared to the *bgl*A gene expressed in *E. coli* M15.

The β-glucosidase of *Thermoanaerobacterium thermosaccharolyticum* DSM 571 worked optimally at 60 °C. The specific activity of the purified enzyme and the kinetic constants $K_m$ and $k_{cat}$ for pNPG hydrolysis were 64 U mg$^{-1}$, 0.64 mM and 67.7 s$^{-1}$, respectively, which confirmed that the enzyme has higher catalytic efficiency, and pNPG has higher affinity and specificity to the active site (*Pei et al., 2012*). Furthermore, *Amouri & Gargouri (2006)* reported kinetics of pNPG hydrolysis by β-glucosidase of *Stachybotrys* sp. at 50 °C. The specific activity was 78 U mg$^{-1}$ and $K_m$ 0.2 mM, which highlighted that our β-glucosidase was kinetically highly efficient.

Kinetics of CMC hydrolysis by endoglucanase was also determined by applying the direct fit method (Fig. 7B). The specific activity of intracellular endoglucanase for CMC hydrolysis by *T. maritima* gene expressed in *E. coli* at 90 °C, pH 4.0 was 16.10 U mg$^{-1}$. The Michaelis Menten constant ($K_m$) for CMC was 9.64 mg ml$^{-1}$.

## Structural basis for thermostability of β-glucosidase from *T. maritima*

Like thermophilic endoglucanases, structural stability of β-glucosidase from *T. maritima* could also be contributed by ionic interactions. More ionic interactions are found in the thermophilic β-glucosidases than in the mesophilic β-glucosidases. Some of these ionic interactions are common, located at corresponding positions in the structure, in both the thermophilic β-glucosidases from *T. maritima* (PDB codes: 1UZ1, *Vincent et al., 2004*;

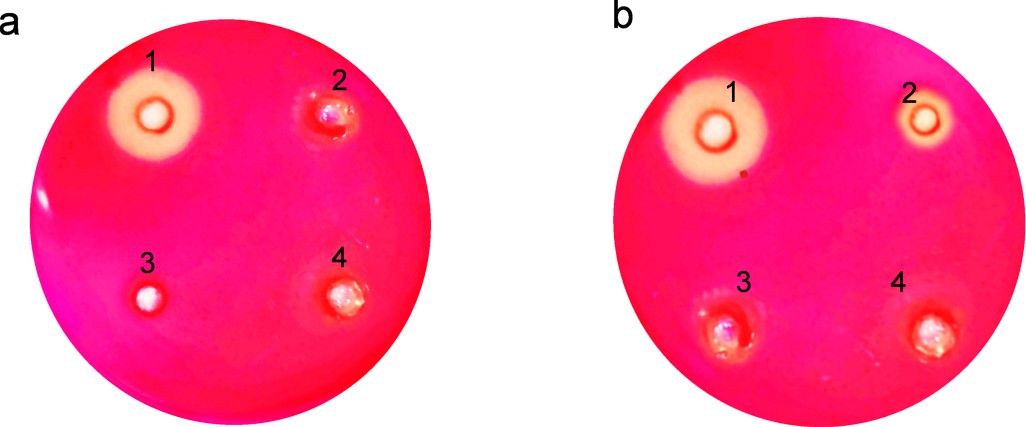

**Figure 4 Activity analysis of endoglucanase expressed in *E. coli* on CMC agar plates.** Enzyme activity was determined by negative staining with Congo red. (A) Detection of endoglucanase activity in intracellular and extracellular extracts. 1. BL21(DE3)-ENDO-intracellular extract, 2. BL21(DE3)-ENDO-extracellular extract, 3. BL21(DE3)-intracellular extract; 4. BL21(DE3)-extracellular extract. (B) Detection of endoglucanase activity in soluble and insoluble content of intracellular extracts. 1. BL21 (DE3)-ENDO-soluble extract; 2. BL21(DE3)-ENDO-insoluble extract, 3. BL21(DE3)-soluble extract, 4. BL21(DE3)-ENDO-insoluble extract                             

2CBU, *Gloster, Madsen & Davies, 2006*; 2WBG, *Aguilar-Moncayo et al., 2009*) and *T. neapolitana* (PDB code: 5IDI; *Kulkarni et al., 2017*) and mesophilic β-glucosidases from *Lactococcus lactis* (PDB code: 1PBG, *Wiesmann et al., 1995*), and *Arabidopsis thaliana* (PDB code: 7F3A, *Horikoshi et al., 2021*). Importantly, there are ionic interactions that are only observed in the case of β-glucosidases from thermophilic bacteria but are missing in β-glucosidases from mesophiles (Fig. 8, Table S2). Thus, higher thermostability of β-glucosidase from *T. maritima* could be due to the presence of 'additional' ionic interactions not found in mesophilic β-glucosidases.

## Structural basis for thermostability of endoglucanase from *T. maritima*

The structural analysis of endoglucanase from *T. maritima* (PDB codes: 3AMH and 3AMM; *Cheng et al., 2011*) revealed that the protein predominantly (>59%) consists of beta (β)-sheets. Notably, a high content (>55%) of β-sheets is also found in endoglucanases from other organisms *Thermococcus sp. 2319x1* (PDB code: 7S8K), *Pyrococcus furiosus* (PDB code: 3VGI, *Kim, Kataoka & Ishikawa, 2012*), *Streptomyces lividans* (PDB code: 2NLR, *Sulzenbacher et al., 1999*) and *Streptomyces sp. 11AG8* (PDB code: 1OA4, *Sandgren et al., 2003*) as given in Table S3. In endoglucanases, β-sheets are arranged as convex outer β-sheets (Sheet A) and concave inner β-sheets (Sheet B). It has been suggested that outer β-sheets (Sheet A) are important for stability as they play a structural role, whereas the inner β-sheets (Sheet B) harbour the active site (*Cheng et al., 2011*). Importantly, the number of residues involved in the formation of Sheet A are more in thermophilic endoglucanases as compared to that in mesophilic endoglucanases (Fig. S2, Table S4). Longer β-sheets (Sheet A) in thermophilic endoglucanases may stabilize the structure and could contribute to thermal stability of the protein. Furthermore, the number of ionic interactions found in thermophilic endoglucanases is also more than that in mesophilic endoglucanases.

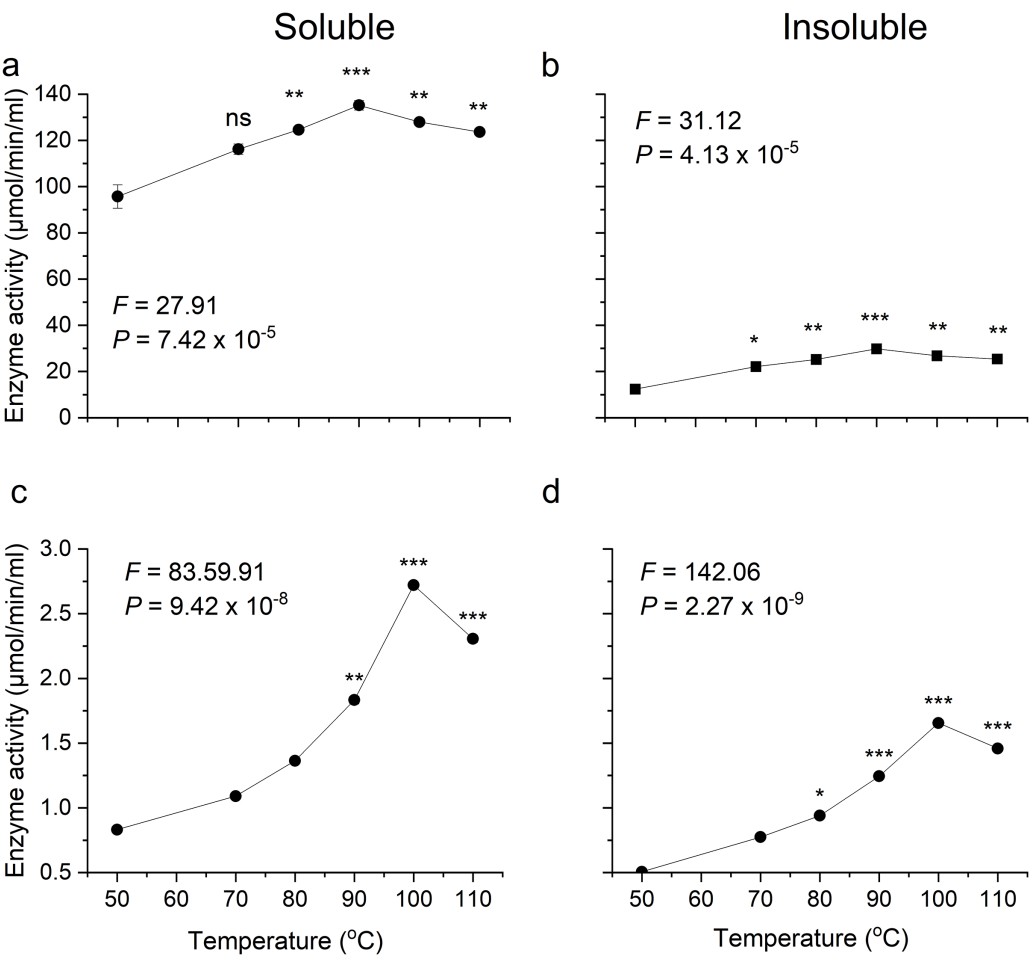

**Figure 5 Activity analysis of enzymes expressed in *E. coli* over a range of temperatures.** (A) and (B) show activity of recombinant β-glucosidase in two separate fractions, soluble (A) and insoluble (B). While (C) and (D) show the activity of endoglucanase enzyme produced by *E. coli* in soluble (C) and insoluble (D) fractions. Data points are the means ± SD of three independent experiments. Significance differences in the data values are marked by asterisks (Tukey's HSD, $P = 0.05$). Significance codes: ***, **, * at $P = 0.001$, $0.01$ and $0.05$, respectively.

Importantly, some ionic interactions are only found in thermophilic endoglucanases and are absent at structurally equivalent positions in mesophilic endoglucanases. Thus, higher thermostability of endoglucanase from *T. maritima* could also be due to the presence of ionic interactions not found in mesophilic endoglucanases (Fig. 9, Table S5).

## DISCUSSION

This study highlights the valuable potential of chloroplast promoters to provide means for the heterologous expression of proteins at large scale from *E. coli*. Chloroplasts possess a semi-autonomous genetic system, with their own gene expression and regulatory elements, including promoters, terminators, and untranslated regions (UTRs). Expression of transgenes from the plastid genome presents distinct advantages, such as elevated expression levels, the absence of epigenetic interactions, and significantly enhanced transgene confinement arising from the maternal inheritance, a salient feature that makes

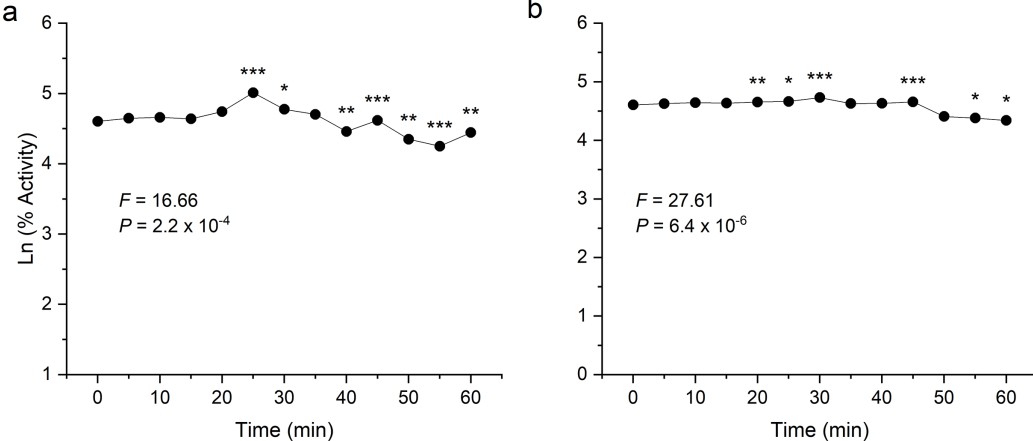

**Figure 6 Thermostability analysis of enzymes expressed in *E. coli*.** (A) shows the activity of recombinant β-glucosidase. (B) shows the activity of recombinant endoglucanase. Data points are the means ± SD of three independent experiments. Significance differences are marked by asterisks (Tukey's HSD, $P = 0.05$). Significance codes: ***, **, * at $P = 0.001$, $0.01$ and $0.05$, respectively.

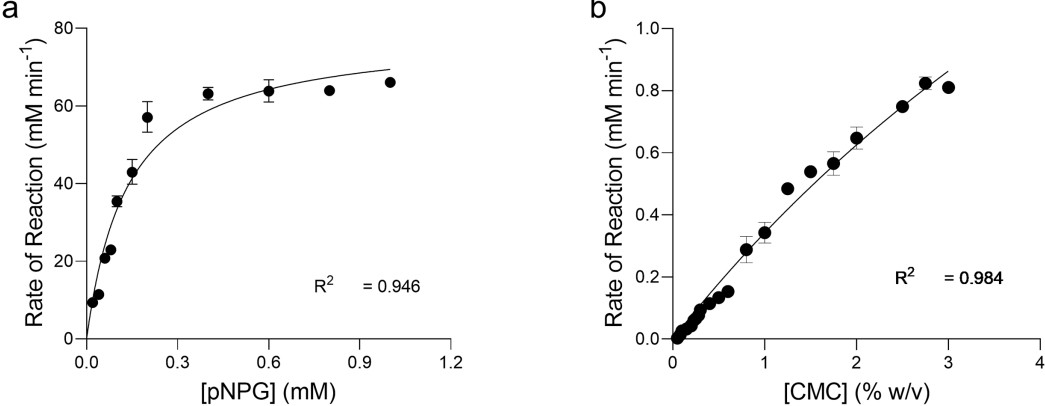

**Figure 7 Enzyme kinetics of pNPG and CMC hydrolysis by recombinant enzymes.** Michaelis-Menten Kinetic ($V_{max}$, $K_m$) constants for the p-nitrophenyl b-D-glucopyranoside (pNPG) hydrolysis and carboxymethyl cellulose (CMC) hydrolysis by recombinant β-glucosidase (A) and endoglucanase (B) expressed in *E. coli* at 90 °C, pH 4.0 and 4.8, respectively were determined by using Direct Fit Model. Data points represent the means ± SD of three experiments.

**Table 1 Kinetics of pNPG and CMC hydrolysis by heterologously-expressed β-glucosidase and endoglucanase in *E. coli* at 90 °C.**

| Properties | β-glucosidase | Endoglucanase |
|---|---|---|
| $V_{max}$ (U mg$^{-1}$ min$^{-1}$) | 283.2 | 16.10 |
| $K_m$ | 0.136 | 96.0 |
| $V_{max}/K_m$ | 2,078 | 0.168 |
| $K_{cat}$ (s$^{-1}$) | 245 | 8.05 |
| $K_{cat}/K_m$ | 1,801 | 0.084 |

**Note:**
Where, units for $K_m$ of pNPG & CMC was mM and mg ml$^{-1}$, respectively.
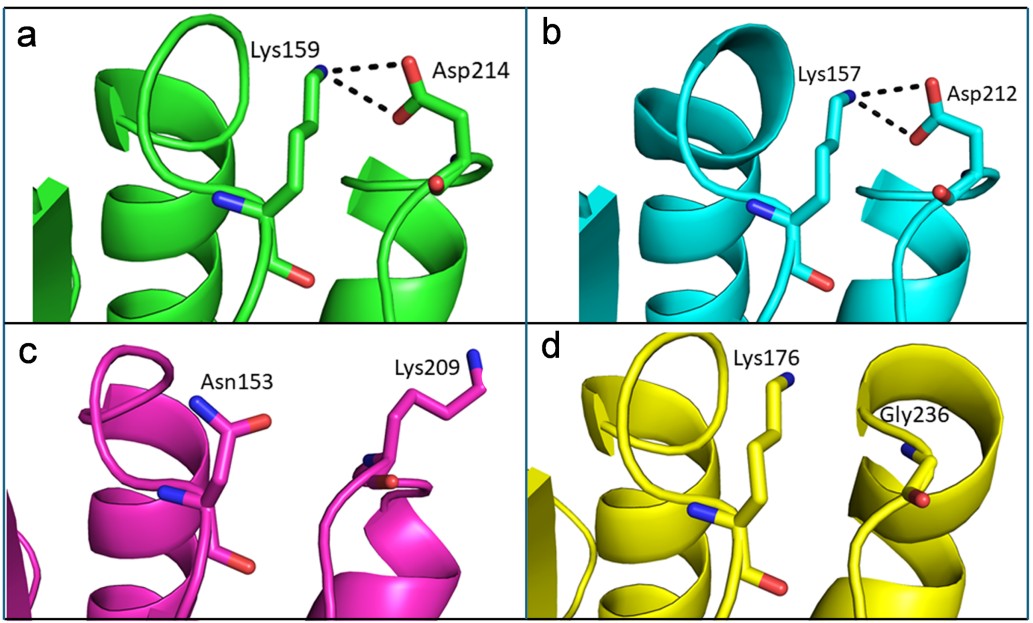

**Figure 8 Differences of ionic interactions between b-glucosidase of thermophilic and mesophilic organisms.** (A and B) Representative ionic interactions (Lys159:Asp214 and Lys157:Asp212; shown with dotted lines) formed between side chains atoms of the acidic and basic residues in beta-glucosidases from thermophilic bacteria (*T. maritima* (1UZ1; A) and *T. neapolitana* (5IDI; B). (C and D) Ionic interactions at the corresponding positions in the structure of beta-glucosidases from mesophilic organisms (*Lactococcus lactis* (1PBG; C), and *Arabidopsis thaliana* (7F3A; D).

transformation of plastids in major field crop species more desirable than conventional approaches (*Ruf, Karcher & Bock, 2007*; *Svab & Maliga, 2007*). Several studies have reported extra-ordinary protein expression levels when transgenes were introduced into higher plant chloroplast genome particularly tobacco (*Ahmad et al., 2012*; *Bock, 2021*; *Daniell et al., 2004*; *Oey et al., 2009*). We determined to combine the ease of transformation and culturing of bacterial system, which would offer utilizing the potential of gene regulatory elements of higher plant chloroplasts for heterologous expression of foreign proteins. During this investigation, we were able to successfully demonstrate the heterologous expression of two widely sought thermotolerant cellulolytic enzymes using a chloroplast transformation cassette in *E. coli*. The enzymes β-glucosidase and endoglucanase from the thermophilic bacterium *Thermotoga maritima* were selected for their efficient cellulose degradation at high temperatures. This characteristic makes them particularly valuable for industrial applications that require enzymatic processing under extreme conditions.

Bacteria, especially *E. coli*, have been extensively utilized for heterologous protein expression due to their low cost, easy transformation, and rapid reproduction (*Kovačević et al., 2024*). *E. coli* constitutes the largest group of bacteria used for this purpose due to the specific properties conferred by its genetic makeup on the products of interest. We developed a plasmid vector with sequences targeting the plastid genome. Despite translation in a different organism, these sequences showed effective protein expression

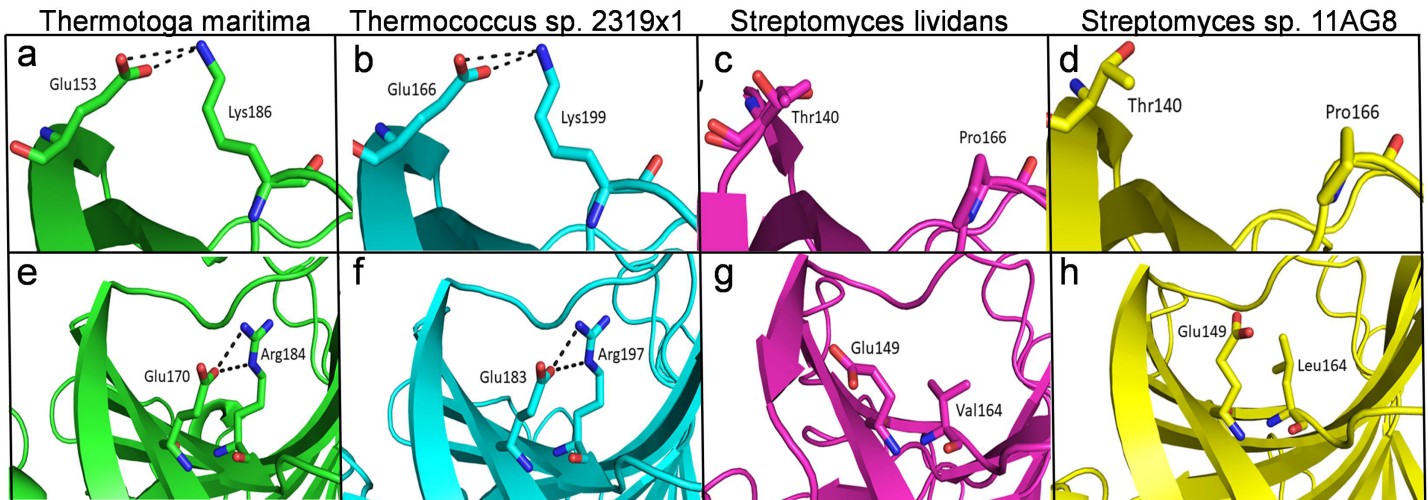

**Figure 9 Differences of ionic interactions between endoglucanases of thermophilic and mesophilic bacteria.** (A, B, E, F) show representative ionic interactions (Glu153:Lys186 and Glu166:Lys199; Glu170:Arg184 and Glu183:Arg197 shown with dotted lines) formed between side chains atoms of acidic and basic residues in endoglucanases from *T. maritima* and *Thermococcus* sp. 2,319 × 1. (C, D, G, H) Ionic interactions at the corresponding positions in the structure of endoglucanases from mesophilic bacteria.

attributed to the evolutionary homology between bacteria and green plastids (*Harada et al., 2024*).

The *E. coli* BL21(DE3) strain is particularly favoured for its ease of culture and high protein yield, making it ideal for industrial applications where enzyme isolation and purification are paramount. The robust genetic toolkit available for *E. coli* simplifies the introduction and expression of foreign genes, enhancing the system's usability. Economically, *E. coli* expression systems are cost-effective compared to other systems, providing an economical option for protein production.

Furthermore, the versatility, scalability, and capacity of *E. coli* to produce properly folded proteins make it a preferred choice for a wide range of protein expression needs. Factors such as the choice of expression vectors, promoters, competent strains, codon optimization, inducer concentration, and optimization of culture conditions play crucial roles in the successful expression of recombinant proteins in *E. coli*.

However, the *E. coli* system is limited to intracellular expression. Studies have shown that all six cellulases could be expressed in tobacco, indicating that the lack of exocellulases in bacterial extracts was not due to issues with the coding sequences. Analysis by western blot revealed protein bands corresponding to soluble enzymes, yet exocellulases were absent, likely degraded rather than misfolded in inclusion bodies (*Klinger, Fischer & Commandeur, 2015*).

Previous research noted that exocellulase E3 was successfully expressed in *E. coli* (*Jeoh, Wilson & Walker, 2002*; *Zhang, Irwin & Wilson, 2000*). Disulfide bond formation, essential for functional exocellulase, was confirmed for E3 (*Rouvinen et al., 1990*), but when expressed in *Streptomyces lividans*, E6 lacked free SH groups (*Zhang, Irwin & Wilson, 2000*). The failure to form disulfide bonds effectively in the cytosolic environment of *E. coli* could account for the observed lack of exocellulase expression (*De Marco, 2009*). A

potential solution to enhance expression involves introducing genes for extracellular secretion pathways, which are not native to *E. coli* (*Pouresmaeil & Azizi-Dargahlou, 2023*).

We utilized the well-established Prrn promoter, derived from the 16S rRNA gene, to express two cellulases in *E. coli*. This promoter is known for its strength in the chloroplast genomes of higher plants, facilitating high levels of heterologous protein expression. Several studies have reported substantial expression levels using the Prrn promoter, underscoring its efficacy in transgenic applications (*Liao et al., 2023*; *Zhang et al., 2015*). For instance, overexpression of the phage lytic protein plyGBS in tobacco chloroplasts using the Prrn promoter resulted in significant accumulation, demonstrating the promoter's potential (*Oey et al., 2009*; *Ruhlman et al., 2010*).

The constitutive plastid ribosomal RNA promoter (Prrn) employed in this study is well-documented for its recognition and activity in *E. coli* (*Drechsel & Bock, 2011*). In our vector, these plastid elements show high protein expression levels in the total soluble protein (TSP) of the cell content indicating potential of these specifically chosen genetic elements in the plasmid vector. Our results correspond with the results of *Petersen & Bock (2011)*, where they expressed endoglucanase in tobacco chloroplasts and observed a maximum of approximately 40% TSP, using the same genetic elements we have used in this work. Similarly, studies conducted by *Castiglia et al. (2016)*, reported expression of recombinant β-glucosidase to be 75.6 ± 4.3% and 60.9 ± 2.1% TSP in tobacco chloroplasts. They achieved high transgene expression by linking the strong plastid 16S rRNA promoter (Prrn) to the coding regions, along with the 5′ translation control region (5′-TCR) that comprises the 5′-UTR and the initial 42 N-terminal nucleotides (DB) of the *atpB* or *rbcL* open reading frames. In addition to the promoter, the possible high expression level obtained in our study could also be explained by the structural integrity of transcribed mRNA which is provided by the 5′ and 3′ UTR elements as used in different protein expression studies for chloroplast (*Eberhard, Drapier & Wollman, 2002*; *Kang et al., 2004*).

Many industries currently use chemicals or high temperatures at the pretreatment stage to convert lignocellulosic biomass to more accessible material for subsequent fermentation (*Petersen & Bock, 2011*). Therefore, cost-effective production of cellulolytic enzymes capable to withstand high temperatures to degrade lignocellulosic biomass is highly desired to make lignocellulosic biofuels commercially feasible. We expressed cellulases from a thermophilic bacterium, *T. maritima*, to work out whether heterologous expression impacts their thermotolerance (*Castiglia et al., 2016*). Efforts on improving thermostability and enzymatic activity of cellulases using site directed mutagenesis and optimum pH requirements of mutants did not yield significant differences compared to unmodified strains (*Xue et al., 2021*). We were able to produce these enzymes *via* heterologous expression in *E. coli*. Our results show that the recombinant enzymes did not show any loss of thermotolerance in their optimal temperature requirements. *Rahman et al. (2002)* reported optimal temperature of endoglucanase enzyme from *T. maritima*, of size 38 kDa was found to be 90 °C while its optimal pH was 6.6, whereas our data shows that endoglucanase had its optimal activity at 100 °C. In case of β-glucosidase our results conclude that it was highly thermoactive showing its optimal activity at 90 °C, which were

in line with earlier studies (*Mehmood et al., 2014*). The results indicated that our cellulases are expressed in an active state.

Our structural analysis of the enzymes (endoglucanase and β-glucosidase) from *T. maritima* suggests that ionic interactions could contribute to thermostability of the enzymes. The role of ionic interactions in enhancing thermostability of various enzymes has been documented previously (*Charbonneau & Beauregard, 2013*; *Ishak et al., 2020*; *Lee et al., 2014*; *Vetriani et al., 1998*). Furthermore, longer β-sheets in endoglucanase from *T. maritima* than that found in homologous endoglucanases from mesophilic organisms also seem to contribute to thermostability of the enzyme. Altogether, the findings deepen our understanding of thermostability of endoglucanase and β-glucosidase from *T. maritima* and pave the way for improving thermostability of their mesophilic counterparts through protein engineering approaches in future studies.

Overall, this work provides a comprehensive analysis of the advantages provided by the specific genetic elements intended for plastid based heterologous protein expression. These elements have proven exceptionally useful in non-native bacterial expression system to provide cost effective thermotolerant cell-wall degrading enzymes that can become a means of steady supply of these enzymes to be used in production of valuable bioproducts from agricultural wastes. Furthermore, this research work would be further expanded to higher organisms for plastid based heterologous protein expression to compare the advantages of each system and provide the best option to choose from.

## Conclusion and future perspectives

In conclusion, this research achieved successful heterologous expression of β-glucosidase and endoglucanase enzymes, native to the hyper-thermophilic bacterium *Thermotoga maritima*, in the industrially favoured *E. coli* BL21 using chloroplast genetic machinery elements. By employing a combination of a constitutive 16S rRNA promoter, *rps16* terminator (Trps16), and 5′ UTRs, we attained high expression levels for endoglucanase and β-glucosidase. Notably, the recombinant enzymes maintained their high thermotolerance, indicating the potential for producing thermotolerant cellulolytic enzymes at high levels. This high-level expression of thermotolerant cellulases holds promise for making the bioconversion of lignocellulosic biomass commercially feasible.

While this study successfully focused on heterologous expression in *E. coli*, these findings provide a foundation for future applications in plant-based systems. The *E. coli* expression system serves as a critical preliminary step in demonstrating the feasibility of utilizing chloroplast genetic elements to achieve high-level expression of thermostable enzymes. Building upon these results, future research can explore the extension of this approach to plant systems, where plastid transformation has already shown significant potential in enhancing transgene expression, reducing enzyme production costs, and enabling large-scale production. Notably, a parallel study is currently being conducted to apply this strategy to plant systems, and the initial results are promising. However, this research is ongoing, and the corresponding manuscript has yet to be submitted for publication.

Future research will focus on optimizing expression levels, maximizing enzyme activity under varying conditions, and exploring diverse applications of these enzymes in biotechnology and industrial settings. This study represents a significant step toward harnessing the potential of thermophilic enzymes for biotechnological applications. By leveraging the unique capabilities of plastid genetic elements, this study would lead to unlock the potential for cost-effective and sustainable production of various products, contributing to the advancement in developing sustainable biomanufacturing platforms to produce specially biomolecules for diverse industrial applications.

## ACKNOWLEDGEMENTS

We are thankful to Dr Hamid Rashid for his help in determining enzyme kinetics.

### Funding

The authors received no funding for this work.

### Competing Interests

Muhammad Aamer Mehmood is an Academic Editor for PeerJ.

### Author Contributions

- Ayesha Siddiqui conceived and designed the experiments, performed the experiments, authored or reviewed drafts of the article, and approved the final draft.
- Muhammad Mudassar Iqbal conceived and designed the experiments, performed the experiments, prepared figures and/or tables, authored or reviewed drafts of the article, and approved the final draft.
- Asad Ali conceived and designed the experiments, performed the experiments, prepared figures and/or tables, authored or reviewed drafts of the article, and approved the final draft.
- Iqra Fatima conceived and designed the experiments, authored or reviewed drafts of the article, and approved the final draft.
- Hazrat Ali conceived and designed the experiments, prepared figures and/or tables, authored or reviewed drafts of the article, and approved the final draft.
- Aamir Shehzad conceived and designed the experiments, performed the experiments, prepared figures and/or tables, authored or reviewed drafts of the article, and approved the final draft.
- Sameer H. Qari conceived and designed the experiments, performed the experiments, authored or reviewed drafts of the article, and approved the final draft.
- Ghulam Raza analyzed the data, authored or reviewed drafts of the article, and approved the final draft.
- Muhammad Aamer Mehmood analyzed the data, authored or reviewed drafts of the article, and approved the final draft.

 - Peter J. Nixon conceived and designed the experiments, analyzed the data, authored or reviewed drafts of the article, and approved the final draft.
 - Niaz Ahmad conceived and designed the experiments, analyzed the data, prepared figures and/or tables, authored or reviewed drafts of the article, and approved the final draft.

## Data Availability

Raw data are provided as a Supplemental File.

## Supplemental Information

Supplemental information for this article can be found online at http://dx.doi.org/10.7717/peerj.18616#supplemental-information.

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
