# Peer review of "Harnessing the potential of chloroplast-derived expression elements for enhanced production of cellulases in Escherichia coli"

_PeerJ, doi:10.7717/peerj.18616_

## Round 0.1 · original submission · Major Revisions

Please address the concerns and questions from the reviewers and provide a revised version along with a rebuttal letter.

Reviewer 1 ·

Basic reporting

BASIC REPORTING
The writing quality of the entire work in general is low, ambiguous and requires the support of an English language editor. It requires improvement of the syntax, the structure, to improve the clarity of the work. The table where the primers used in the work are listed is irrelevant, if they do not explain how they were used. The image of the enzymatic activity (Figure 4) is not necessary. It would have been more relevant to determine the kinetic constants of both purified enzymes and show them.

Experimental design

EXPERIMENTAL DESIGN
The contribution of the work to knowledge is not clear, since chloroplast promoters have already been previously studied and are functional in bacteria, as they mention. The procedure for the design of the expression cassette, which may be the most relevant part of the work, is not described since they do not indicate how they optimized the genes, what changes were made to the genes to adapt them to the recipient strain, they do not indicate where to locate the sequences of the genes used, they do not indicate how they used the designed primers or what the purpose of the design was, they conclude that they managed to obtain a high concentration of enzymes but do not show numerical values ​​and do not compare with other works, they indicate that the system can be used for the production of thermostable enzymes, however this characteristic is inherent to the primary structure of the enzyme and to the type of microorganism to which it belongs, and not to the recipient microorganism.

Validity of the findings

VALIDITY OF THE FINDINGS
The results are valid, the necessary data are provided, however, I consider that important determinations were missing, such as the kinetic parameters of the enzyme, its thermoresistance, its affinity for different substrates… etc.

Additional comments

General comments
I think that the writing and structure of the work needs polishing, being more concise in discussing the results and conclusions, showing and highlighting the truly relevant aspects of the work.

Annotated reviews are not available for download in order to protect the identity of reviewers who chose to remain anonymous.

Reviewer 2 ·

Basic reporting

1. Rewrite lines 55-56, and 60-62 for better understanding.
2. In lines 91-92, please explain the extreme conditions for the enzymes.
3. Please include a transition statement between lines 92-93.
4. What is meant by transportation fuel production in line 102?
5. In line 103, you mentioned fungi are historically preferred and then in line 104, you mentioned bacterial cellulase is preferred, so which one is better?
6. In line 113, why do you have to enhance enzyme activity at elevated temperatures, please provide a brief description.
7. Rewrite lines 116-122.
8. In lines 128-129, do the enzyme Endoglucanase and betaglucosidase belong to the cellulases or xylanases groups?
9. Could you explain why you chose the pNGP assay for betaglucosidase in line 348 and the CMC plate assay for endoglucanase in line 355 and write a brief description of it?
10. Break down the paragraphs 437-487 into multiple paragraphs for better understanding.
11. Merge lines 418-420 as it is kind of repetitive.

Experimental design

no comment

Validity of the findings

no comment

Reviewer 3 ·

Basic reporting

1. Rewrite lines 65-67, 89-80, and 98-100 for the clear understanding.
2. In lines 98-100, explain how the use of cellulases in the recycling of paper will in related to bioenergy production.
3. Justify the difference in the statements in lines 103-106.
4. Add a connecting line between 122-123.
5. Please include some background information about the enzymes in the first few paragraphs and explain in detail about the Thermotoga martima in lines 128-131.
6. Define all abbreviations upon first use in the manuscript, for example, PCR, LB-agar, etc.
7. In line 348, how does the pNPG assay work, and why it is chosen?
8. Lines 434-436 are a little complex and could be made more concise.

Experimental design

no comment

Validity of the findings

no comment

---

## Round 0.2 · accepted · Accept

Thank you for incorporating all the final comments in your last revision; your manuscript has now been accepted by PeerJ!

Reviewer 1 ·

Basic reporting

The quality of writing of the entire work improved considerably. The table where the primers used in the work are listed, explains their use and the kinetic constants of both enzymes were determined.

Experimental design

They explain the interest of the study of chloroplast promoters, and describe the methodology used for the design of the expression cassettes and the optimization of the genes, the changes made to the genes according to the Kazusa codon usage tables, indicate where the sequences of the genes studied are noted, indicate how they used the designed primers and their purpose, and explain how they determined the level of expression of the enzymes.

Validity of the findings

The results are valid, the necessary data are provided, and important determinations were made such as the kinetic parameters of the enzyme, its heat resistance.

Additional comments

The relevant aspects of the work are highlighted, I consider that the reviewers' suggestions were addressed, and the work can be accepted.

Reviewer 2 ·

Basic reporting

no comment

Experimental design

no comment

Validity of the findings

no comment

Additional comments

no comment

Reviewer 3 ·

Basic reporting

NO COMMENT

Experimental design

NO COMMENT

Validity of the findings

NO COMMENT

Additional comments

NO COMMENT